# Do Preparation Techniques Transform the Metabolite Profile of Platelet-Rich Plasma?

**DOI:** 10.3390/bioengineering12070774

**Published:** 2025-07-17

**Authors:** Bilge Başak Fidan, Emine Koç, Emine Çiftçi Özotuk, Ozan Kaplan, Mustafa Çelebier, Feza Korkusuz

**Affiliations:** 1Department of Bioengineering, Graduate School of Science and Engineering, Hacettepe University, Ankara 06800, Türkiye; bilgefidan98@gmail.com; 2Department of Analytical Chemistry, Faculty of Pharmacy, Hacettepe University, Ankara 06230, Türkiye; emikocc25@gmail.com (E.K.); ozankaplan@hacettepe.edu.tr (O.K.); mcelebier@gmail.com (M.Ç.); 3Department of Sports Medicine, Faculty of Medicine, Hacettepe University, Ankara 06230, Türkiye; emineeciftci@gmail.com

**Keywords:** platelet-rich plasma (PRP), untargeted metabolomics, Q-TOF LC/MS

## Abstract

Background: Platelet-rich plasma (PRP) is a widely used therapeutic product in musculoskeletal treatments due to its regenerative and anti-inflammatory properties. However, the lack of standardization in PRP preparation protocols hampers clinical consistency. Methods: In this study, the metabolic profiles of 10 different PRP types were compared using untargeted metabolomics via Q-TOF LC–MS. PRP-G and PRP-S were prepared from six donors to assess inter-individual variability, while the remaining types were obtained from a single donor to isolate the impact of preparation method alone. Multivariate analyses, VIP scores, and pathway enrichment analyses were conducted. Results: PRP formulations exhibited distinct metabolic differences associated with inflammatory signaling, redox homeostasis, steroid metabolism, energy production, and platelet activation. Samples from both single- and multi-donor groups showed high intra-group similarity, indicating that preparation method is a major determinant of PRP’s biochemical composition. Conclusion: Metabolomic profiling reveals that even minor differences in PRP preparation protocols can lead to significant biochemical changes that may affect therapeutic outcomes. This study highlights the need for standardized, indication-specific PRP products and underscores the value of metabolomic analysis in guiding optimal formulation selection in clinical practice.

## 1. Introduction

Platelet-rich plasma (PRP) has become an increasingly preferred anti-inflammatory and regenerative treatment method for the symptomatic treatment of musculoskeletal conditions [1]. Various PRP variations involving different concentrations and activation methods have recently been developed [2]. These include preparation methods of leukocyte-poor PRP (P-PRP), leukocyte-rich PRP (L-PRP), and platelet-rich fibrin (PRF). Platelet-derived growth factor, epidermal growth factor, insulin-like growth factor I, transforming growth factor beta, and angiogenic growth factors such as fibroblast growth factors and vascular endothelial growth factors in PRPs are assumed to decrease inflammation and regulate tissue regeneration [3]. They promote cell proliferation, stimulate the formation of new blood vessels, and accelerate tissue healing. The efficacy and stability of PRP can significantly vary depending on its preparation and activation methods, including the leukocyte content [3]. L-PRP is usually preferred in periarticular and tendon applications, whereas P-PRP is more frequently used as an intra-articular injection [4]. Multiple intra-articular P-PRP injections have recently been recommended for the symptomatic treatment of knee osteoarthritis [5].

Metabolomics has long been used in clinical applications to investigate disease mechanisms, monitor treatment responses, and identify diagnostic biomarkers. This approach can be applied to PRP, another biologically active fluid, to identify and compare the biochemical differences among its various formulations. By doing this, metabolomics provides a way to examine how different preparation methods may affect PRPs [6,7,8].

A previous article [9] reported on proteomic-level data of PRPs, while research at the metabolomic level remains limited. That study identified more than 100 canonical pathways with significant overlap between plasma formulations in terms of protein content. Among the most represented pathways were the acute-phase reactants and naturally the molecules of coagulation, as well as the complement systems, highlighting the association with inflammation [9]. In another study, metabolomic analyses were performed on the stability of PRP, which is frequently used in the treatment of OA. These analyses suggested that frozen and reused PRP may not pose toxicological risks, but it may not fully replicate the physiological efficacy of freshly prepared PRP. In light of these findings, it is recommended that PRP be prepared fresh to ensure optimal effectiveness [10]. In another study [11] 21 proteins were found to be significantly different between the cellular fractions of PRP and PRF at the proteomic level. Among these proteins, fibronectin and thrombospondin 1 merged as key molecules involved in tissue repair and neovascularization. Their regenerative effects at the metabolomic level, however, were not presented. Multi-omics has shown that PRP can enhance the biological performance of mesenchymal stem cells by promoting mechanisms such as cell proliferation, adhesion, migration, and signal transduction. This improvement is achieved through cellular metabolic reprogramming facilitated by various growth factors and extracellular vesicles within PRP [6]. The importance of metabolomic analyses furthermore has been highlighted in trauma studies related to platelet transfusion. Metabolomic approaches have identified changes associated with the presence of specific metabolites such as tranexamic acid, fatty acids, and polyamines in patients undergoing platelet transfusion [12]. Analyzing the metabolomics of PRP can therefore help uncover not only its bioactive potential but also the underlying reasons for the differences observed between various formulations—such as variations in platelet content, activation methods, or additive components—since metabolites reflect the end products of cellular and biochemical processes. In this context, our primary research question was whether differences in PRP preparation methods regardless of donor variability led to significant alterations in the metabolomic profiles of PRP. We hypothesized that specific PRP preparation protocols generate distinct metabolite compositions and pathway enrichments, which in turn may underlie differences in their therapeutic potential and clinical applicability.

In this study, we utilized an advanced untargeted metabolomic approach employing quadrupole time-of-flight liquid chromatography–mass spectrometry (Q-TOF LC–MS) technology to systematically investigate and compare the metabolite profiles of two activated PRP preparations, PRP-G and PRP-S, derived from six healthy donors. We additionally analyzed eight distinct PRP preparations (A-PRP, TFF-PRP, L-PRP, P-PRP, C_L-PRP, C_P-PRP, PRP-P, and B7-PRP) obtained from a single donor to assess intra-individual variability and the biochemical diversity introduced by different preparation methods. This comprehensive analysis aimed to elucidate how specific PRP preparation protocols influence its biological composition and therapeutic potential by identifying key metabolites and relevant metabolic pathways.

## 2. Materials and Methods

### 2.1. Design

We designed a systematic, multi-stage and multidisciplinary experimental study utilizing advanced analytical methodologies to explore metabolomic variations among different PRP preparations. Untargeted metabolomic analyses were conducted using LC–MS Q-TOF technology to characterize activated PRP-G and PRP-S prepared from six healthy donors. Metabolomic diversity among eight distinct PRP types (A-PRP, TFF-PRP, L-PRP, P-PRP, C_L-PRP, C_P-PRP, PRP-P, and B7-PRP) from a single donor was furthermore assessed. Independent variables included PRP preparation methods and metabolomic analysis parameters, whereas dependent variables were defined as the resulting metabolomic profiles and related biochemical changes due to the variations in the preparation procedures. This study was approved by the Hacettepe University Non-Invasive Clinical Research Ethics Committee (approval date 20 September 2022, approval no 2022/14-57).

Six healthy male individuals aged 18–24 years volunteered for the first stage of the study. Inclusion criteria required participants to be male and free from medication use, which was considered an exclusion criterion (Table 1). Blood count results (Appendix A) and descriptive statistics of the participants of this stage of research were obtained. One participant who was receiving chemotherapy was replaced with another volunteer due to exclusion criteria and abnormal blood cell count.

The second part of the study explored intra-individual variability by analyzing eight distinct PRP types prepared from blood samples obtained from a healthy 62-year-old male donor. This donor had no known chronic diseases, medication use, or smoking habits, and had a height of 172 cm and weight of 104 kg, representing the male obese patient profile. PRP preparations were carried out according to protocols provided by the manufacturer’s instructions of each of the eight commercially available PRP kits used.

The clear distinction between multiple-donor analysis and the single-donor variability assessment ensured comprehensive coverage and clarity regarding the metabolomic diversity introduced by different PRP preparation methods.

### 2.2. PRP Preparation Procedure

PRP-G and PRP-S (Quantix, Milan, Italy) samples were prepared using two commercial kits specifically designed for producing mechanically activated PRP, incorporating specialized tube compositions and activation mechanisms. The remaining eight PRP types were obtained using various commercial systems that differ in terms of activation status, leukocyte content, additives, and centrifugation protocols. This method-based diversity, rooted in the commercial nature of the kits, was intentionally selected to reflect the heterogeneity observed in clinical practice and to enable comparative evaluation at the metabolomic level. The detailed specifications of each PRP preparation method, including centrifugation type and conditions (RPM and duration), processing capacity, yield, tube material, activation status, and leukocyte and platelet content are summarized in Table 2.
Stage 1: Blood Collection

Venous blood samples were collected from the antecubital vein under sterile conditions using standard venipuncture techniques. Blood samples were immediately transferred into specialized tubes corresponding to the specific PRP preparation method used. A CE-marked and FDA-approved separation kit (Neogenesis PRP, Seoul, Republic of Korea) was used to prepare L-PRP and P-PRP. An anticoagulant (0.13% natrium citricum, PPS, MediPac, Köningswinter, Germany) was used to prepare these conventional PRPs. Another CE-marked conventional kit (CellPhi PRP, Istanbul, Türkiye) was used to prepare C_L-PRP and C_P-PRP, which were used to validate the former separation kit. To activate the PRP, the PRP-G (Quantix IDRIA G, Milan, Italy) and the PRP-S (Quantix IDRIA S+, Milan, Italy) preparation kits that contained beads were preferred. For the chemical activation of the PRP, A-PRP, PRM-PRP, and B7-PRP (MSK kits, London, UK) systems were used. The A-PRP, PRM-PRP, and B7-PRP kits contained hyaluronan and vitamin B7 for activation of the platelets. The TFF-PRP preparation kit (Beijing Manson Technology Co., Ltd., Beijing, China) was chosen to validate the B7-PRP (Table 2).Stage 2: Centrifugation

Collected blood samples were centrifuged according to manufacturer-specific protocols using precisely defined parameters (speed, duration, temperature) optimized for each PRP preparation method. Centrifugation separated the blood into distinct layers, including platelet-poor plasma, PRP, and red blood cells (Table 2).Stage 3: Fractionation and PRP Collection

Following centrifugation, the desired PRP fractions were carefully isolated. The platelet-rich layer was selectively collected with necessary adjustments to platelet concentrations performed depending on the PRP type and intended clinical use. Specific protocols adhered strictly to producers’ manuals, ensuring the integrity and consistency of each prepared PRP sample.

### 2.3. Q-TOF LC–MS Analysis

PRP samples were first subjected to protein precipitation using methanol for metabolomic analysis. From each PRP type, 0.1 mL of sample was transferred to microcentrifuge tubes and mixed with 0.2 mL of methanol–water (95:5 *v*/*v*). The mixture was vortexed and centrifuged at 10,000 rpm for 10 min at +4 °C in a refrigerated centrifuge (Hettich Universal 320 R, Tuttlingen, Germany). Following centrifugation, 0.1 mL of the supernatant was collected and evaporated to dryness using a vacuum concentrator set to 9 °C (CentriVap, Labconco Co., Kansas City, MO, USA).

The dried residues were reconstituted in 0.1 mL acetonitrile–water (1:1, *v*/*v*), vortexed thoroughly, and centrifuged again to remove particulates. From the resulting clear supernatant, 0.2 mL was transferred to individual LC–MS vials for single injections. Pooled samples were prepared by combining 0.05 mL of each replicate within a PRP group to ensure analytical consistency. Quality control samples were additionally generated by pooling 0.05 mL from each individual sample across all groups.

Chromatographic separation was carried out on a Q-TOF LC–MS system (Agilent 6530 Tech. Santa Clara, CA, USA) equipped with a C18 column (2.1 × 110 mm, 2.5 µm particle size, XBridge, Waters, Milford, MA, USA). The mobile phases consisted of water with 0.1% formic acid (A) and acetonitrile with 0.1% formic acid (B). The gradient elution program was as follows: 90% A (0–1 min), 65% A (1–4 min), 10% A (4–11 min), and 90% A (11–14 min), held at 90% A until 20 min. The flow rate was 0.4 mL/min, the injection volume was 5 µL, and the column temperature was maintained at 35 °C.

Mass spectrometric detection was performed in negative ion mode over a scan range of *m*/*z* 75–1200. Sample injections were randomized to minimize batch effects. Mobile phase blanks and quality control samples were injected every twelve runs throughout the analysis sequence to monitor system stability and reproducibility.

### 2.4. Data Analysis

Metabolomic data were processed using MZmine 2.53, a widely used software platform for untargeted metabolomics. The workflow included peak detection, chromatographic alignment, deconvolution, and peak grouping. Resulting peak lists were exported to Microsoft Excel for organization and downstream statistical analysis. Multivariate statistical analyses were conducted using MetaboAnalyst 6.0. Principal component analysis (PCA) was employed to visualize the clustering patterns and detect group-wise separations based on metabolite composition. Partial least squares discriminant analysis (PLS-DA) was subsequently used to identify key discriminative metabolites, with variable importance in projection (VIP) scores > 1 considered indicative of a significant contribution to group differentiation. Heatmaps were generated to visualize relative intensities of these discriminative metabolites across sample groups. Metabolite annotation was initially performed using major public databases—the Human Metabolome Database (HMDB), Kyoto Encyclopedia of Genes and Genomes (KEGG), and METLIN—by matching accurate mass, isotope patterns, and retention time predictions. Where possible, identification was further confirmed by comparison with in-house reference standards run under identical chromatographic and mass spectrometry (MS) conditions. Enrichment analysis was performed to contextualize the findings biologically. In this analysis, significantly altered metabolites represented in the heatmap were mapped to relevant biological pathways using MetaboAnalyst 6.0 software. This comprehensive workflow enabled detailed characterization of how PRP preparation techniques modulate biochemical composition and potential clinical functionality.

Two separate raw data files are provided in the Appendix A to ensure transparency and reproducibility. The dataset was deposited in a public repository but is fully accessible through the Appendix A submitted with the manuscript.

## 3. Results

Metabolomic analysis revealed distinct biochemical profiles among the PRP samples. Mechanically stimulated PRP-G and PRP-S samples formed distinct and non-overlapping clusters, indicating substantial differences in their metabolomic profiles (Figure 1). This clear separation reflected that the two mechanically activated PRP types possess unique biochemical compositions. Such separation was critical, as it suggested that even within the category of activated PRP, preparation techniques can introduce metabolomic diversity that may influence their respective biological and clinical effects.

VIP scores quantitatively reflected the contribution of each metabolite to the overall group separation, with higher scores indicating a greater role in distinguishing between sample groups in multivariate models. Metabolites with VIP scores greater than the 1.0 threshold, which is a commonly accepted threshold indicating significant contribution to group separation, are presented (Figure 2). Examples including 1-phosphatidyl-D-myo-inositol, OPC6-CoA, cyclopentanone, and phosphatidylinositol-3,4,5-triphosphate played a key role in differentiating PRP-G from PRP-S (Figure 3). Red and blue boxes next to each metabolite indicate relative abundance in PRP-G and PRP-S samples, respectively. Red represents higher levels and blue represents lower levels, allowing visual comparison of metabolite distribution between the two PRP types.

The heatmap presents the relative abundance of discriminative metabolites in PRP-G and PRP-S samples (Figure 3). The observed variations reflected both individual-specific metabolic profiles and differences introduced by the PRP preparation protocols. The clustering pattern within each PRP group demonstrates the analytical consistency and repeatability of the respective PRP kits despite inter-individual variability, supporting their reliability in producing distinct metabolomic signatures.

Enrichment analysis is a bioinformatic method used to determine whether specific metabolic pathways are statistically overrepresented among a list of significantly altered metabolites, providing insight into potential biological functions or mechanisms (Figure 4). In this study, enrichment analysis was conducted using metabolites that showed a fold change greater than 1.5 between PRP-G and PRP-S groups and were found to be statistically significant (*p* < 0.05) based on unpaired *t*-tests. This approach enabled the identification of metabolic pathways most influenced by the differences in PRP preparation methods.

The PCA score plot revealed distinct clustering among the eight PRP types prepared from the same individual, indicating that preparation technique alone can generate substantial metabolic diversity even in the absence of inter-individual variation (Figure 5). The clear group-wise separation highlights that the biochemical composition of PRP was strongly influenced by the specific commercial kit and protocol used, rather than donor-related factors.

VIP scores were calculated to identify the metabolites driving these separations (Figure 6). These scores rank metabolites according to their contribution to the differentiation between PRP types. Metabolites with VIP scores greater than 1 are considered highly influential in separating PRP formulations, offering valuable insight into how specific preparation methods shape the metabolic fingerprint of PRP.

A heatmap was generated to visualize the concentration patterns of significant metabolites across the eight PRP types (Figure 7). The relative abundance of these metabolites varied widely among the different formulations, with certain profiles—such as those of A-PRP and TFF-PRP—showing elevated levels of specific compounds. These patterns further underscore the non-equivalence of PRP types, even when derived from the same individual, and reflect both the chemical selectivity of the kits and the influence of processing protocols.

Enrichment analysis was performed using metabolites that showed statistically significant differences (*p* < 0.05) and fold changes greater than 1.5 between groups. Several pathways were significantly enriched, including porphyrin metabolism, steroid hormone biosynthesis, and energy-related pathways such as beta-alanine and propanoate metabolism (Figure 8). These results suggest that different PRP preparations may exert distinct biological effects ranging from inflammatory modulation to muscle and tissue metabolism support, depending on their metabolomic composition.

These findings together confirm that standardization in PRP preparation is essential for achieving consistent therapeutic outcomes and that formulation-specific selection may be required based on clinical objectives such as tissue regeneration or anti-inflammatory response.

## 4. Discussion

Despite its broad clinical application in musculoskeletal disorders, PRP’s bioactive effects remain highly variable due to non-standardized preparation techniques, making it essential to evaluate the molecular consequences of these procedural differences [13,14]. Metabolomics can provide a powerful analytical method to characterize PRP formulations at the molecular level, enabling detailed insight into their biochemical compositions beyond the growth factor content alone [15]. This approach allows clinicians and researchers to identify specific metabolite-level differences linked to various preparation methods, enhancing the understanding of PRP’s functional variability and guiding more tailored and mechanism-driven clinical applications. In this study, an untargeted metabolomic approach using Q-TOF LC–MS technology was used to systematically analyze PRP formulations prepared by different commercial methods. Heatmaps and enrichment analyses, which provided functional insights beyond mere statistical separation, are shown to interpret the results from a clinical perspective effectively. Inter-individual and preparation method variability significantly influenced metabolomic profiles, while PCA plots clearly illustrated distinct clustering of PRP groups. Analytical outcomes bridged the gap between raw data and biological relevance, facilitating interpretation in terms of inflammatory potential, regenerative capacity, and therapeutic consistency.

Enrichment analysis identified several metabolic pathways significantly altered between PRP-G and PRP-S preparations, both of which use microbeads and gel for activation. Pathways including steroidogenesis, carnitine biosynthesis, selenoamino acid metabolism, inositol phosphate metabolism, biotin metabolism, and the citric acid cycle were prominently enriched, indicating substantial biochemical divergence arising from different activation methods. Steroidogenesis is critical for synthesizing hormones involved in inflammation regulation, immune modulation, and tissue healing [16,17]. Inositol phosphate metabolism contributes to intracellular signaling mechanisms, particularly calcium signaling, essential for platelet activation and aggregation, thus potentially impacting PRP efficacy [18,19,20]. Carnitine biosynthesis plays a pivotal role in energy metabolism and oxidative stress response, which may affect tissue repair processes and cellular protection mechanisms [21,22,23]. The citric acid cycle is an additionally fundamental pathway that is involved in cellular energy production and metabolic regulation [24], influencing overall cellular health and therapeutic outcomes. Significant differences were also observed in metabolic pathways such as homocysteine degradation, aspartate metabolism, butyrate metabolism, the mitochondrial electron transport chain, and riboflavin metabolism, which were among the most enriched between the PRP-G and PRP-S formulations. Although both PRP-G and PRP-S were generated using commercial kits from the same manufacturer, they were distinguished by differences in centrifugation protocols, activation processes, and tube composition. These procedural variations led to significant metabolic differences, as indicated by the enriched pathways observed. Such metabolomic variability may subsequently influence distinct biological responses and biological effects.

Importantly, the enriched pathways identified through this analysis are not based on the exclusive presence or absence of specific metabolites in either PRP-G or PRP-S, but rather on substantial quantitative differences in metabolite abundance. These metabolites are generally present in both PRP types. Their concentrations may differ dramatically, indicating that certain metabolites are nearly absent in one formulation relative to the other.

VIP analysis identified key discriminative metabolites between PRP-G and PRP-S, shown in heatmaps. Relative abundance patterns demonstrated the biological differences between the formulations. Though pathway enrichment indicated relevant metabolic pathways, it did not reveal which specific compounds were responsible. Focusing on individual metabolites is crucial to determine which PRP type suits particular therapeutic aims, as identifying these metabolites links PRP composition to biological outcomes. Notably, 1-phosphatidyl-D-myo-inositol and other phosphatidylinositols were substantially more abundant in PRP-G. These metabolites are critical components of the inositol phosphate signaling pathway, which regulates intracellular calcium release and is essential for platelet activation and cell signaling [25,26]. Their elevated presence in PRP-G suggests a potentially enhanced capacity for cellular activation and signal transduction, which may be advantageous in applications requiring rapid tissue response or regeneration [27]. Metabolites involved in oxidative stress regulation such as Se-methylselenocysteine [28] and N′-formylkynurenine were also elevated in PRP-G. This supported the synthesis of antioxidant selenoproteins [29] and Se-methylselenocysteine, which was in line with the changes in phosphatidylinositols. The simultaneous elevation of metabolites involved in inositol signaling and redox regulation highlighted PRP-G’s dual biochemical profile, combining strong signaling activation potential with oxidative stress resilience. This may make PRP-G particularly attractive for regenerative therapies targeting acute injury prone to oxidative damage.

In contrast, PRP-S was characterized by relatively higher levels of metabolites such as biocytin, a biotin-conjugated lysine derivative known to function as a coenzyme in several carboxylase-dependent metabolic pathways, including fatty acid synthesis, amino acid metabolism, and gluconeogenesis [30,31,32]. Biotinylated platelet formulations [32] notably have been actively studied in transfusion medicine, demonstrating that biotin and its derivatives are not only metabolically relevant but also functionally retained by platelets in vivo, influencing their post-transfusion physiology and metabolic behavior. The presence of such metabolites in PRP-S may reflect a biochemical profile conducive to metabolic stability and structural maintenance, rather than acute pro-regenerative signaling. Accordingly, PRP-S might be more appropriate for therapeutic applications requiring gradual tissue support and sustained metabolic balance, such as in chronic musculoskeletal conditions.

While these metabolomic distinctions offer valuable mechanistic insights into the potential functional profiles of PRP-G and PRP-S, their biological relevance should ideally be validated through in vitro or in vivo studies. Observing cellular or tissue-level responses to these two formulations under controlled experimental conditions would help confirm whether the metabolic signatures reported here align with measurable therapeutic outcomes. Such integrated evaluations are expected to strengthen the translational significance of our findings and guide formulation-specific clinical applications.

The second part of this study aimed to isolate the influence of preparation techniques by eliminating inter-individual biological variability. Eight distinct PRP formulations were prepared from the same participant, allowing a focused comparison of how various commercial PRP preparation kits affect the metabolic composition of the final product. The PCA revealed clear separation among these eight PRP types. This underscored the strong methodological impact of kit-specific protocols such as centrifugation parameters, activation agents, and fractionation strategies on the biochemical profiles of PRP. The observed clustering indicates that even subtle differences in preparation methodology can lead to profound changes in the metabolomic landscape of PRP, which may translate into distinct therapeutic behaviors. Enrichment analysis presented a comprehensive view of the metabolic pathways differentially affected across the eight distinct PRP types. Observed differences were related to intrinsic compositional diversity among the formulations themselves. Pathways such as alpha-linolenic and linoleic acid metabolism, catecholamine biosynthesis, and arginine and proline metabolism were prominent. Each pathway was contributing to a distinct biochemical signature for different PRP types. These pathway-level distinctions suggest that certain formulations may be more suitable for specific therapeutic applications, ranging from pro-resolving inflammation to enhancing tissue remodeling, depending on their dominant metabolic traits. This functional divergence was further supported by the concordance between PCA and heatmap analysis, where formulations like A-PRP and TFF-PRP not only form separate clusters but also exhibit differential abundance of key metabolites such as bilirubin, uracil, testosterone glucuronide, arachidonic acid, and galactosylsphingosine. A-PRP, for example, which appears isolated in the upper-left quadrant of the PCA graph, was characterized by markedly elevated levels of bilirubin, arachidonic acid, coproporphyrinogen, and creatine. The identified metabolites—such as arachidonic acid, coproporphyrinogen, and creatine—have previously been implicated in oxidative stress, inflammatory signaling, heme metabolism, and energy mobilization pathways in osteoarthritic contexts, as supported by both our prior work and independent studies [33,34,35,36], reinforcing the biological plausibility of the metabolic shifts observed in PRP formulations.

Elevated bilirubin levels have been associated with reduced oxidative stress and inflammation in various conditions, including metabolic syndrome and diabetes [37]. Arachidonic acid is a polyunsaturated omega-6 fatty acid integral to cell membrane phospholipids. Upon cellular activation or injury, phospholipase A2 releases arachidonic acid, which is then metabolized by cyclooxygenase and lipoxygenase into eicosanoids, including prostaglandins, thromboxanes, and leukotrienes. These metabolites were key mediators of inflammation, pain, and vascular responses [38,39]. Coproporphyrinogen III [40] is an intermediate molecule in the heme biosynthetic pathway. It undergoes oxidative decarboxylation by coproporphyrinogen oxidase to form protoporphyrinogen IX, which is a precursor to heme. Heme is essential for various biological functions, including oxygen transport, electron transfer, and enzymatic reactions [41].

A study [42] reported that heme-induced platelet activation involves the generation of reactive oxygen species, and ferroptosis was an iron-dependent form of regulated cell death characterized by lipid peroxidation. That study demonstrated that exposure to heme led to increased reactive oxygen species production in platelets, resulting in oxidative stress and subsequent activation of ferroptotic pathways. This process contributes to platelet activation and aggregation, highlighting a novel mechanism by which heme can influence platelet function and potentially exacerbate thrombotic events in hemolytic conditions. Taken together, these findings suggest that A-PRP enriched in bilirubin, arachidonic acid, coproporphyrinogen III and creatine exhibits a metabolomic profile consistent with high redox buffering capacity, early-phase inflammatory activation, and enhanced bioenergetic support. Such a formulation may be ideal in acute regenerative scenarios, where rapid cellular mobilization, controlled inflammation, and metabolic activation are required.

The close proximity of clustering in both PCA and heatmap-based metabolite patterns of C_L-PRP and C_P-PRP suggested that they share a similar biochemical background, likely due to comparable leukocyte and platelet compositions or analogous activation protocols. Subtle differences in metabolites such as creatine, maleic acid, or testosterone glucuronide, however, may still reflect functional distinctions between them. Minor metabolic variations between C_L-PRP and C_P-PRP may not cause clinically meaningful differences. Notably, testosterone glucuronide levels appear slightly higher in C_P-PRP suggesting a modest elevation in androgen-associated metabolic signaling [43,44]. Additionally, minor variations in creatine and maleic acid concentrations may point to distinctions in energy buffering and redox regulation. This finding correlates with prior findings [45,46] indicating that platelet metabolic activity during storage is influenced by shifts in glycolytic and tricarboxylic acid cycle intermediates. Such metabolic shifts may be relevant in osteoarthritic microenvironments, where PRP efficacy is thought to depend on meeting increased energy demands and managing oxidative stress [47]. Their reproducibility across replicates suggested that even closely related PRP types may possess distinct biochemical profiles, although such metabolic shifts do not result in clear separation in PCA clustering. Targeted evaluation in well-controlled therapeutic settings of PRP is therefore strongly suggested. C_L-PRP, which shows slightly higher levels of creatine and maleic acid, may be more appropriate for treatments requiring long-term energy support and redox balance, as in late-stage osteoarthritis. In contrast, C_P-PRP may be better suited for applications where mild androgenic or anabolic stimulation is beneficial with relatively elevated testosterone glucuronide, such as in the early cartilage regeneration phases.

The comparison between L-PRP and P-PRP demonstrated discrete clustering between the two formulations, indicating consistent divergence in their global metabolic fingerprints. Metabolites such as L-tyrosine, D-glucuronic acid, and nicotinamide riboside, illustrated in the heatmap, revealed distinct differences. L-tyrosine, a precursor for catecholamines, is significantly elevated in L-PRP and has been associated with stress response pathways [48,49]. This elevation can be preferred for applications where immune cell activation and tissue remodeling are desirable. D-glucuronic acid, another discriminant metabolite, is also more abundant in L-PRP and contributes to hyaluronic acid biosynthesis [50], which can be used in OA treatment [51]. Nicotinamide riboside, a precursor of NAD+, is known as anti-inflammatory and regenerative [52]. P-PRP, in contrast, presented lower levels of these inflammatory or remodeling-associated metabolites, favoring a biochemical profile more consistent with controlled inflammation and matrix homeostasis. P-PRP is often better tolerated in intra-articular applications like early-stage osteoarthritis where a low-inflammatory and matrix-preserving environment is expected [53,54]. These findings reinforce the concept of tailoring PRP formulations to clinical needs. L-PRP may be more appropriate for musculoskeletal injuries requiring robust tissue remodeling, while P-PRP may offer advantages in chronic degenerative conditions with low-grade inflammation. Our results are in accordance with those of [54], while [53] reported no significant difference between L-PRP and P-PRP in terms of long-term efficacy.

TFF-PRP presented elevated levels of carnosine [55], testosterone glucuronide [56], L-valine [57], and docosahexaenoic acid [58] metabolites associated with antioxidant activity [59,60], tissue remodeling [61,62], and anabolic signaling. This suggested that TFF-PRP may support restoration of the anabolic–catabolic balance in degenerative joint environment [63] and that this formulation may be particularly suitable for early intervention in osteoarthritic conditions, where enhancing tissue repair while mitigating oxidative and inflammatory damage is critical. This dual capacity makes TFF-PRP a promising candidate for applications requiring controlled, yet active tissue regeneration.

B7-PRP and PRM-PRP occupy central positions in the PCA plot clustering between more metabolically distinct PRP types, such as TFF-PRP, A-PRP, and P-PRP. This intermediate localization suggested a relatively balanced and less polarized metabolic phenotype. B7-PRP and PRM-PRP contained moderate levels of various metabolites, including testosterone glucuronide, L-valine, carnosine, and docosahexaenoic acid without showing strong enrichment or depletion patterns. Such a broad, yet non-specialized biochemical signature implies that these formulations may not be tailored to target specific pathophysiological features such as intense inflammation or regeneration. Instead, their balanced metabolic composition might render them suitable for general-purpose applications or maintenance-phase interventions where a stable and moderately bioactive environment is desired (Table 3).

### 4.1. Reproducibility of the Kits

Despite the data being derived from a single donor, the clear clustering patterns observed in the PCA (Figure 5) and the consistent metabolite distributions across technical replicates underscore the methodological reproducibility of the evaluated PRP preparation kits. The metabolomic consistency across replicate samples within each kit suggests that the separation protocols yield metabolically stable and reproducible formulations. This is particularly important for clinical and research applications, where batch-to-batch consistency directly impacts therapeutic predictability and scientific validity. Hence, the distinct metabolic phenotypes of each PRP type observed in this study can be confidently attributed to the kit-specific processing characteristics rather than random variation.

### 4.2. Limitations and Future Perspectives

While this study provides a comprehensive metabolomic characterization of various PRP formulations using an untargeted Q-TOF LC–MS approach, certain limitations must be acknowledged. First, the number of platelets and leucocytes after PRP preparations was not counted, as the amount of the samples was limited to only metabolomic analysis. The number of platelets is clinically important; however, this may not alter metabolomic outcomes. Second, although several key metabolites were matched with authentic standards, the untargeted nature of the method inherently limits definitive compound identification for all discriminative features. Incorporating complementary analytical techniques—such as MS/MS-based spectral confirmation, targeted LC–MS/MS validation, or nuclear magnetic spectroscopy—would enhance structural confidence and quantitation. Second, it is also important to note that the datasets comparing PRP-G vs. PRP-S and the eight PRP types were acquired at different time points using separate LC–MS runs. For this reason, they were not analyzed within the same dataset to avoid technical variation caused by batch effects. Additionally, it was not feasible to detect the full metabolome in a single injection due to the nature of untargeted LC–MS analysis. Only 3000–5000 features are captured on average, and 700–1000 can be annotated and reliably compared. The absence of overlapping VIP metabolites between figures therefore reflects the inherent limitations of untargeted metabolomics, batch separation, and the complexity of pathway coverage. Future studies using targeted approaches may help reveal common metabolites across PRP types more consistently. Third, a key limitation of this study was that the intra-individual comparison of eight different PRP types was performed using blood from a single overweight male donor with no symptoms of OA, who represented the male obese patient profile. While this limits the generalizability of the findings, it also eliminates inter-individual biological variability, allowing us to isolate the effects of preparation methods on metabolomic profiles. Since all samples were derived from the same blood source, the observed differences can be confidently attributed to the PRP kits themselves. Supporting this approach, the comparison of PRP-G and PRP-S samples obtained from six volunteers demonstrated strong intra-group clustering and clear inter-group separation, indicating that the preparation protocol is the dominant factor influencing PRP composition. The lack of direct pairwise functional comparisons among the eight PRP types, however, limits the ability to draw definitive conclusions about clinically relevant metabolic differences. Future studies involving broader and more diverse populations, along with direct comparative and functional assessments, will therefore be crucial. Such studies will contribute to a deeper understanding of PRP formulations and guide personalized treatment strategies. Fourth, the metabolomic profiles were evaluated in the absence of functional or cellular bioassays such as in vitro application of these PRPs, which limits translational interpretation. Moreover, the biological activity of PRP is not solely determined by its metabolite content: bioactivity is also profoundly influenced by protein components—especially growth factors, cytokines, and extracellular vesicles—which were not assessed in this study. Future investigations integrating metabolomics with proteomic, transcriptomic, and functional assays will be essential for a more holistic understanding of PRP formulations. Additionally, evaluating sex- and age-related variability, long-term storage effects, and formulation-specific outcomes in disease-relevant models such as osteoarthritis and tendinopathies will be critical to developing evidence-based, personalized PRP selection strategies.

## 5. Conclusions

This study demonstrates that distinct PRP formulations exhibit metabolically diverse profiles that need to be considered for clinical applications. Activation strategies of PRP are reinforcing the notion that a universal strategy is unlikely to be optimal across all clinical conditions. Instead, PRP selection should be informed by the specific therapeutic objective, such as the stage of tissue inflammation or intra- or peri-articular repair and regeneration. Relevant storage considerations for multiple applications also deserve critical attention. Our findings support a more rational and personalized approach to PRP-based interventions in musculoskeletal medicine by identifying formulation-specific metabolites associated with inflammation modulation, tissue regeneration, and redox balance. Tailoring PRP choice based on biochemical composition may enhance clinical outcomes and advance the precision application of orthobiological therapies.

## Figures and Tables

**Figure 1 bioengineering-12-00774-f001:**
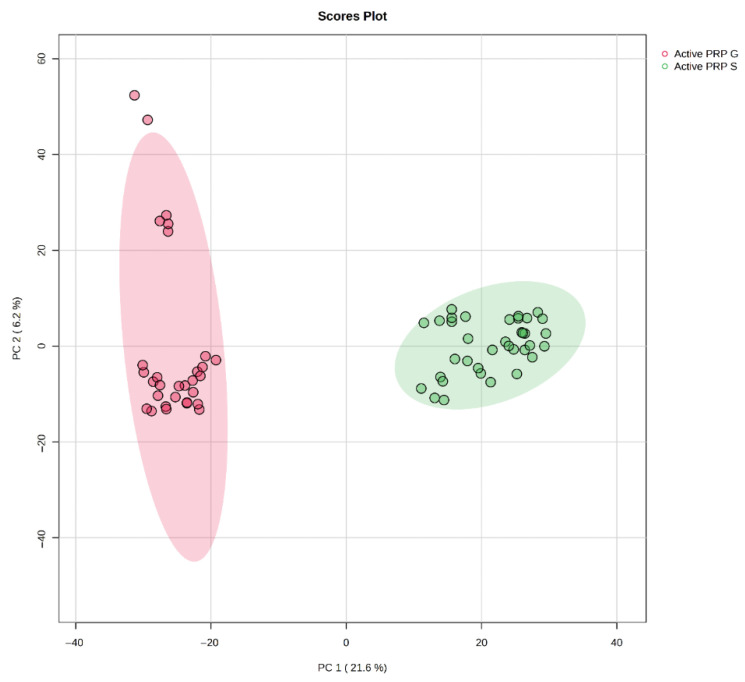
PCA score plot showing distinct clustering of PRP-G and PRP-S samples.

**Figure 2 bioengineering-12-00774-f002:**
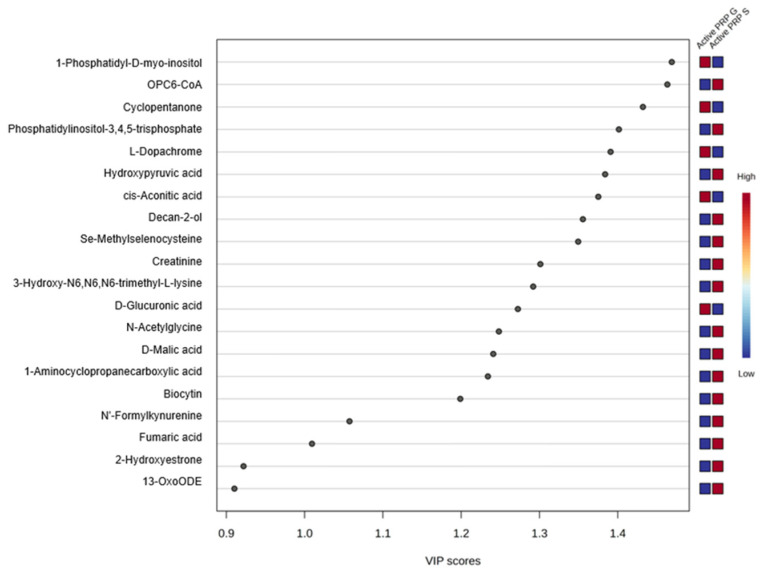
VIP score plot displaying the top metabolites contributing to the separation between PRP-G and PRP-S.

**Figure 3 bioengineering-12-00774-f003:**
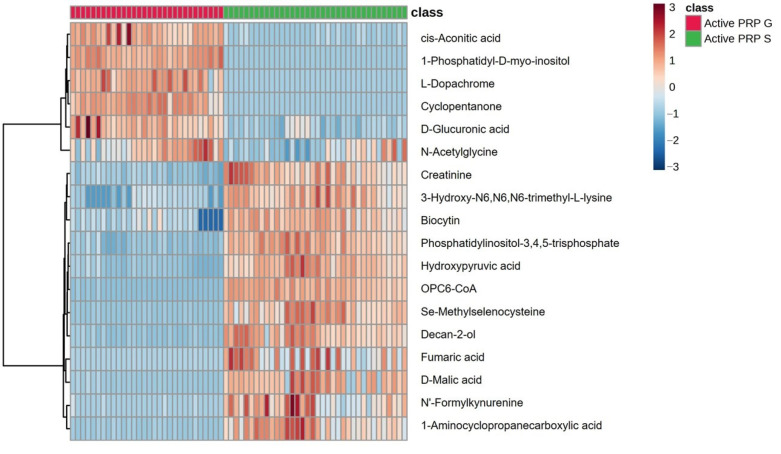
Heatmap (VIP > 1) illustrating the relative abundance of key discriminative metabolites between PRP-G and PRP-S, with red indicating higher levels and blue indicating lower levels.

**Figure 4 bioengineering-12-00774-f004:**
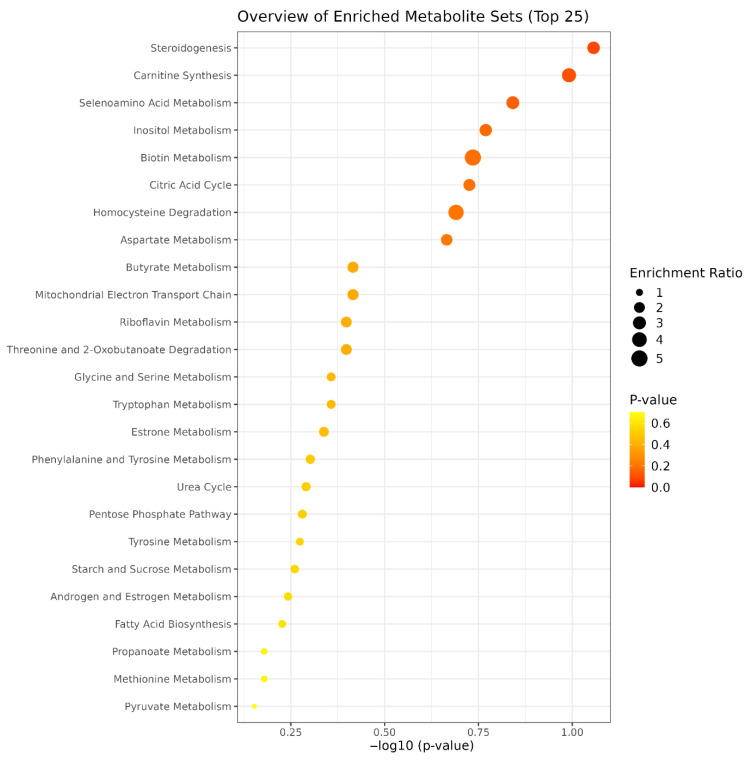
Enrichment analysis of significantly altered metabolites between PRP-G and PRP-S. Pathways are ranked based on statistical significance (−log10 *p*-value), with dot size indicating the enrichment ratio and color representing *p*-value intensity.

**Figure 5 bioengineering-12-00774-f005:**
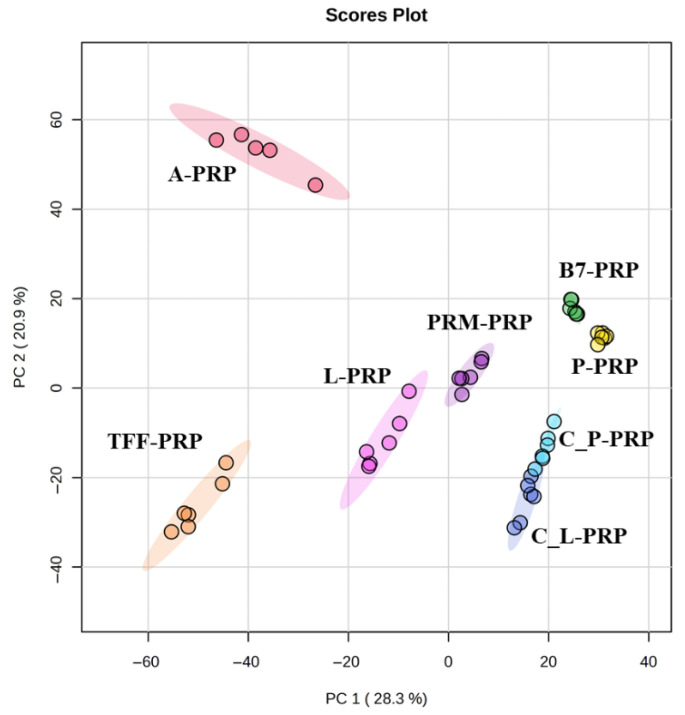
PCA score plot illustrating the separation of eight distinct PRP types prepared from a single donor.

**Figure 6 bioengineering-12-00774-f006:**
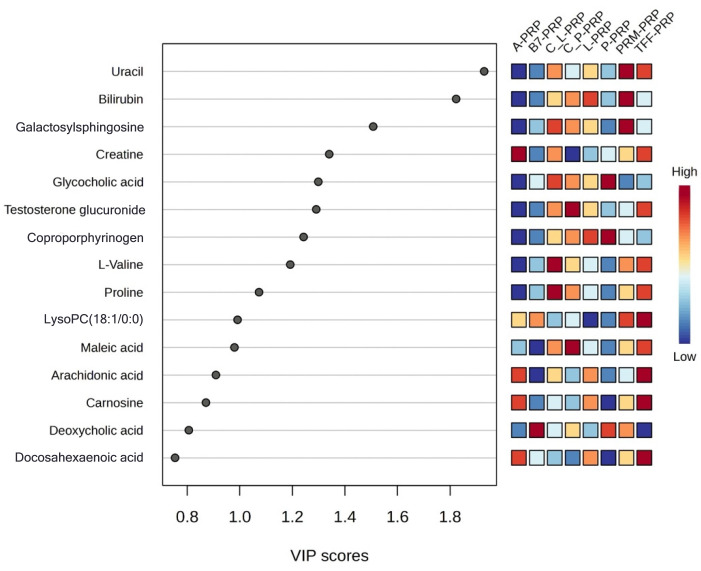
VIP score plot presenting the top metabolites responsible for the separation between different PRP preparation methods.

**Figure 7 bioengineering-12-00774-f007:**
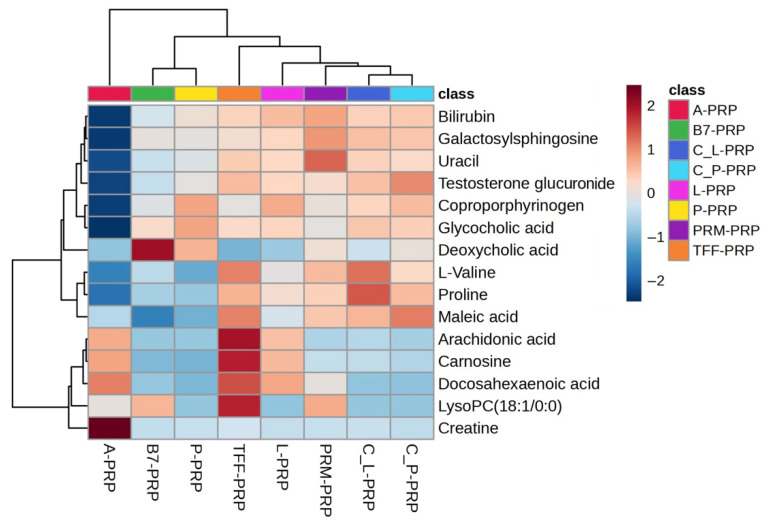
Heatmap (VIP > 1) showing the relative abundance of significant metabolites across eight PRP types. Red and blue indicate higher and lower metabolite levels, respectively.

**Figure 8 bioengineering-12-00774-f008:**
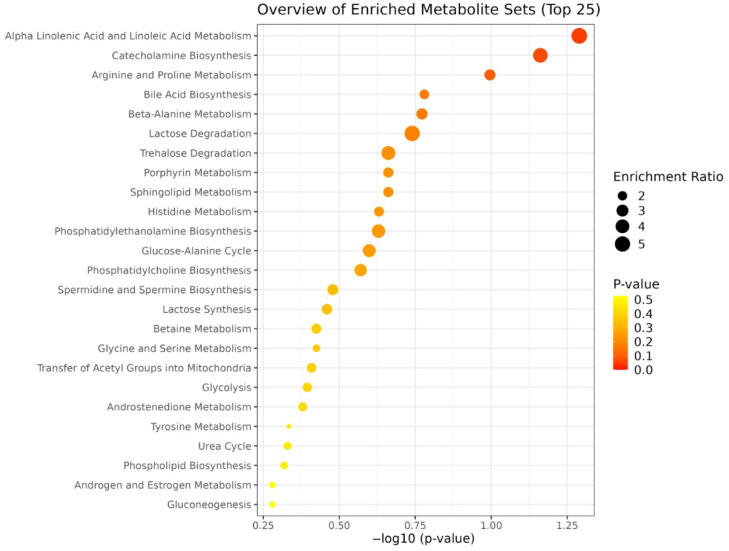
Enrichment analysis comparing different PRP preparation methods.

**Table 1 bioengineering-12-00774-t001:** Demographic characteristics of the participants (*n* = 6).

Variables	Mean ± SD or (%)
Gender	Male, 6 (100%)
Age (years)	22.0 ± 1.7
Height (cm)	178.8 ± 6.2
Weight (kg)	81.2 ± 13.2
Body mass index (kg/m^2^)	26.0 ± 5.0
Comorbidities	Present: 1 (16.6%) Absent: 5 (83.4%)
Smoking status	Smoker: 1 (16.6%) Non-smoker: 5 (83.4%)

**Table 2 bioengineering-12-00774-t002:** Technical specifications and processing parameters of PRP preparation kits *.

PRP Type	Kit Name	Centrifugation (Single/Double Spin)	RPM (Revolutions per Minute)	Duration (min)	Yield (mL)	Tube Material	Activated	Leukocyte Count	Erythrocyte Count	Platelet Count
PRP-P	MSK Premium PRP	Single	3500	8	2–3	Gel-based plastic	No	0.04 × 10^3^/μL		
A-PRP	MSK Activated PRP	Single	3000	6	2–3	Gel-based plastic	Yes	0.03 × 10^3^/μL		
B7-PRP	MSK Biotin PRP	Single	3000	6	4–5	Gel-based plastic	Yes	0.01 × 10^3^/μL		
P-PRP	Neogenesis	Single	3100	10	~3	Plastic	No	0		
L-PRP	Neogenesis	Single	3100	10	~4	Plastic	No	6.49 × 10^3^/μL	0.04 × 10^3^/μL	447 × 10^3^/μL
C P-PRP	CellPhi	Single	3400	10	5.5	Plastic	No	0		
C L-PRP	CellPhi	Single	3400	10	5.5	Plastic	No	1.67 × 10^3^/μL		
TFF-PRP	Manson HA-PRP	Single	3500	8	2–3	Gel-based plastic	Yes	0.56 × 10^3^/μL		
PRP-G	Quantix IDRIA G	Double	4150/3100	3/8	~13	Gel-based plastic	Yes	NA	NA	NA
PRP-S	Quantix IDRIA S	Single	4000	10	~13	Glass beads based plastic	Yes	NA	NA	NA

* Leukocyte, erythrocyte, and thrombocyte count in the L-PRP group before and after PRP preparation changed from 6.66 × 10^3^/μL to 6.49 × 10^3^/μL, 4.31 × 10^3^/μL to 0.004 × 10^3^/μL and 230.00 × 10^3^/μL to 447.25 × 10^3^/μL, respectively. The PRP preparation procedure essentially consisted of three main stages.

**Table 3 bioengineering-12-00774-t003:** Summary of distinctive metabolic features, inferred effects, and clinical recommendations for PRP formulations.

PRP Type	Key Metabolites (Heatmap/VIP)	Main Effect/Discussion	Suggested Clinical Application
A-PRP	Bilirubin, arachidonic acid, coproporphyrinogen, creatine	High redox buffering, early-phase inflammatory activation, enhanced energy support	Acute regeneration, rapid cellular mobilization, controlled inflammation
TFF-PRP	Carnosine, testosterone glucuronide, L-valine, DHA	Antioxidant effect, tissue remodeling, anabolic profile	Early osteoarthritis or conditions with oxidative/inflammatory damage
L-PRP	L-tyrosine, D-glucuronic acid, nicotinamide riboside	High immune activation, tissue remodeling, anti-inflammatory potential	Acute musculoskeletal injuries, tissue remodeling scenarios
P-PRP	Low L-tyrosine & D-glucuronic acid, balanced profile	Low inflammatory, matrix-preserving profile	Chronic degenerative conditions, early osteoarthritis, intra-articular uses where minimal inflammation is desired
C_L-PRP	Creatine, maleic acid (slightly elevated)	Sustained energy support, redox balance	Chronic tendon repair, late-stage osteoarthritis
C_P-PRP	Testosterone glucuronide (slightly elevated), creatine	Androgenic/anabolic stimulus, energy support	Early cartilage regeneration, subacute joint injuries
B7-PRP	Moderate levels of various metabolites	Balanced, non-polarized biochemical profile	General-purpose or maintenance-phase applications
PRM-PRP	Moderate levels of various metabolites	Balanced, non-polarized biochemical profile	Supportive, maintenance, or non-targeted application

## Data Availability

Patient data is strictly protected and cannot be available according to the law #6698 entitled “Protection of Personal Data” that came into force on 24 March 2016 and published in the Official Gazette #29677 according to the Turkish State.

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
