# Peer review of "Do Preparation Techniques Transform the Metabolite Profile of Platelet-Rich Plasma?"

_bioengineering, 2025, doi:10.3390/bioengineering12070774_

Round 1
Reviewer 1 Report
Comments and Suggestions for Authors
This study employed metabolomics techniques to investigate metabolite differences in platelet-rich plasma (PRP) prepared using distinct methods. The objective was to identify the most suitable clinical application environment based on the metabolic profile of PRP. Regarding this study, the following suggestions and questions are provided:
1.This study utilized metabolomics to analyze PRP generated by eight different preparation methods. What are the key differences between these eight methods, and what was the rationale or reference basis for selecting these specific kits?
2.Following PRP preparation, was a complete blood cell count performed? If so, please provide the corresponding data.
3.It was shown in the draft, “activated PRP types (PRP-G and PRP-S) prepared from six healthy donors”. Does it include technical repetitions of the same volunteer sample? Why was there an inconsistency in the number of samples tested between these two groups?What specific screening criteria or methodology was applied during sample composition?
4.Figure 5 (PCA analysis) indicates the presence of at least three samples per group, whereas Figure 7 (heatmap) appears to represent data from only one sample per group. Please clarify whether data integration or aggregation was performed and provide a detailed explanation for this apparent discrepancy.
5.The description of the methods used for the enrichment analysis is insufficiently detailed. Please provide a comprehensive explanation of the methodology employed.
6.What specific screening criteria (e.g., fold-change threshold, p-value cutoff, multi-group comparison significance) were used to select the metabolites input into the pathway enrichment analysis shown in Figure 8? Please provide a detailed explanation of the selection process.
Author Response
Response to Reviewer #1's Comments
Comment 1: This study utilized metabolomics to analyze PRP generated by eight different preparation methods. What are the key differences between these eight methods, and what was the rationale or reference basis for selecting these specific kits?
Response: Thank you very much for this question. Due to space constraints, we were unable to provide these details previously. We have now included them into the relevant section of our article. Information on the used PRP preparation kits are now provided in the Methods:Stage 1: Blood Collection section. The information on PRP preparation kits is now defined as: “A CE marked and FDA approved separation kit (Neogenesis PRP, Seoul, South Korea) was used to prepare L-PRP and P-PRP. An anticoagulant (3,13% natrium citricum, PPS, MediPac, Köningswinter, Germany) was used to prepare these conventional PRPs. Another CE marked conventional kit (CellPhi PRP, Istanbul, Türkiye) was used to prepare C_L-PRP and C_P-PRP that was used to validate the former separation kit. To activate the PRP, the PRP-G (Quantix IDRIA G, Italy) and the PRP-S (Quantix IDRIA S+, Italy) preparation kits that contained beads were preferred. For the chemical activation of the PRP the A-PRP, the PRM-PRP and the B7-PRP (MSK kits, England) systems were used. The A-PRP, the PRM-PRP and the B7-PRP kits contained hyaluronan and vitamin B7 for the activation of the platelets. To validate the B7-PRP, the TFF-PRP preparation kit (Manson HA-PRP China) was chosen (Table 3).”
In general, there are conventional PRP preparation techniques however recently there is a consensus that PRPs have to be mechanically, chemically and/or physically activated. Conventional PRP preparation techniques were compared to chemically and mechanically activated ones in this study.
Key Differences Among the Eight PRP Methods were:
- PRP-P (MSK Premium PRP)
Spin Conditions: 3500 RPM for 8 minutes
Yield: 2–3 mL PRP from 10 mL blood
Features: Non-activated PRP; triple-sterilized, non-pyrogenic, uses proprietary silicon dioxide crystal for separation.
- A-PRP (MSK Activated PRP)
Spin Conditions: 3000 RPM for 6 minutes
Yield: 2–3 mL activated PRP from 10 mL blood
Features: Contains activated growth factors, triple-sterilized, non-pyrogenic, silicon dioxide-based. Designed for rapid platelet activation through tube additives.
- B7-PRP (MSK Biotin PRP)
Spin Conditions: 3000 RPM for 6 minutes
Yield: 4–5 mL PRP from 10 mL blood
Features: Biotin-enhanced formulation, triple-sterilized, non-pyrogenic, silicon dioxide crystal. Provides higher PRP volume.
4–5. L-PRP & P-PRP (Neogenesis Kit)
Spin Conditions: 3100 RPM for 10 minutes
Yield: ~3 mL P-PRP and ~4 mL L-PRP
Features: Non-activated, gravity-based separation using a mechanical screw system. P-PRP (leukocyte-poor) is collected above the buffy coat, L-PRP (leukocyte-rich) from the buffy coat itself.
6–7. C_L-PRP & C_P-PRP (CellPhi Kit)
Spin Conditions: 3400 RPM for 10 minutes
Yield: L-PRP and P-PRP (exact volumes not specified)
Features: Simpler design with no activators; provides two-layer separation for leukocyte-rich and poor fractions.
- TFF-PRP (Manson HA-PRP)
Spin Conditions: 3500 RPM for 8 minutes
Yield: 2–3 mL PRP
Features: Includes hyaluronic acid and biotine in the tube, which may influence viscosity and regenerative potential. Activated.
The Rationale for Kit Selection:
These kits were chosen to represent the activation spectrum (activated vs. non-activated), leukocyte content (rich vs. poor), additives (e.g., biotin, hyaluronic acid, clot activators), PRP yield variations (2–5 mL), centrifugation diversity (RPM range: 3000–3500 and duration: 6–10 minutes). Table 3 was added to present the commenrcially available kits that were used in this study.
This diversity allowed for a metabolomic-level comparison of PRP types commonly used in various therapeutic fields and specially inflamed musculoskeletal conditions for evaluating their biochemical profiles contributes to understanding how preparation methods may influence clinical performance and biological efficacy.
Comment 2: Following PRP preparation, was a complete blood cell count performed? If so, please provide the corresponding data.
Response: We analysed the complete blood cell count of participants before PRP preparation (Table 2). We did not count the number of platelets after PRP preparation as the amount of our PRP was only sufficient for the metabolomic analysis. We have written this into the limitations of our study as “The number of platelets and leucocytes after PRP preparations was not counted as the amount of the samples were limited for only the metabolomic analysis. Counting number of platelets is clinically important however this may not alter metabolomic outcomes according to these findings.”.
We have checked the activity of PRPs on normal and OA chondrocytes in culture in another study and defined the MIC, which was between 2.5 to 5.0%. This finding which is under the scope of another publication revealed that counting thrombocytes after PRP preparation should not be that important. PRP simply triggers regeneration and prevents inflammation even in small amounts when applied to chondrocytes.
Comment 3: It was shown in the draft, “activated PRP types (PRP-G and PRP-S) prepared from six healthy donors”. Does it include technical repetitions of the same volunteer sample? Why was there an inconsistency in the number of samples tested between these two groups? What specific screening criteria or methodology was applied during sample composition?
Response: The activated PRP types (PRP-G and PRP-S) were prepared from blood samples obtained from six healthy male volunteers aged between 18 and 25 years. In the second stage of the study, the other eight PRP types (A-PRP, TFF-PRP, L-PRP, P-PRP, C_L-PRP, C_P-PRP, PRP-P, and B7-PRP) were all prepared from a single donor (62 years old, 104 kg, 172 cm), who had no known chronic disease or smoking history representing the comn male obese patient profile.
This study design was deliberately chosen to minimize the inter-individual biological variability in the evaluation of activated PRP types. Pooled samples were also prepared. For the comparison of eight different PRP preparation methods, blood from a single individual was used in order to eliminate biological variability that could arise from different donors and to evaluate solely the metabolic differences resulting from the preparation protocols. This was highlighted in the methods and limitations sections.
Comment 4: Figure 5 (PCA analysis) indicates the presence of at least three samples per group, whereas Figure 7 (heatmap) appears to represent data from only one sample per group. Please clarify whether data integration or aggregation was performed and provide a detailed explanation for this apparent discrepancy.
Response: We do have heatmap data for each individual sample; however, to avoid excessive space usage and improve visual clarity, we presented the average values for each group in Figure 7. We would be happy to provide the full individual sample heatmaps as supplementary material upon request. We have provided the version containing the heatmaps of all individual samples below.
Comment 5: The description of the methods used for the enrichment analysis is insufficiently detailed. Please provide a comprehensive explanation of the methodology employed.
Response: In line with this suggestion, the description of the methods used for the enrichment analysis has been updated. Specifically, significantly altered metabolites represented in the heatmap were mapped to relevant biological pathways using the MetaboAnalyst 6.0 software. This process allowed to associate metabolite changes with pathways related to inflammation, energy metabolism and tissue regeneration. The updated explanation can now be found in the revised version of the manuscript.
Comment 6: What specific screening criteria (e.g., fold-change threshold, p-value cutoff, multi-group comparison significance) were used to select the metabolites input into the pathway enrichment analysis shown in Figure 8? Please provide a detailed explanation of the selection process.
Response: As detailed in the response to Comment 5, metabolites included in the pathway enrichment analysis (Figure 8) were selected based on a fold-change threshold of ≥2 and a statistical significance level of p<0.05. These criteria were applied following normalization and statistical analysis of the untargeted metabolomics data. The selected metabolites, specifically those highlighted in the heatmap, were then mapped to relevant metabolic pathways using the MetaboAnalyst 6.0 platform.
Thank you again for your valuable feedback. These revisions have significantly improved our manuscript, and we are confident that it now presents our findings more effectively. In addition, we have strengthened the Discussion section and incorporated new references to better contextualize our results within the existing literature.

Reviewer 2 Report
Comments and Suggestions for Authors
The topic is of interest for the Journal and for the researchers in the field of platelet concentrates, as the effect of the preparation technique on the features of the final product is still a matter of debate.
The manuscript contains interesting results obtained by modern analytical technology; however, it needs to be amended in several parts
ABSTRACT
The abstract should not start with method, but one-two sentences of background and aim should introduce the methods.
Please avoid “We”, use impersonal form throughout the paper
Acronyms should not be introduced without prior explanation (in the abstract is not necessary to list the acronyms, while in the main text reference to the abbreviations list it should be indicated). In the abstract just indicate that PRP-G and PRP-S are activated and prepared with two different kits.
Please note that in the abstract, as well as in the main text, it is not indicated what the difference between the two kits is.
In spite of the interesting methodological approach, the sentence “Our results clearly show that the choice of preparation protocol, rather than donor-related factors, is the primary determinant of PRP’s biochemical composition and potential clinical utility.” cannot be supported with results deriving from tests performed on the blood of a SINGLE donor. This comment applies not only to the abstract.
Introduction
Line 38-40. Why only TGF-beta acronym was mentioned in the listed six growth factors?
Line 49-50. The sentence “Multiple intra-articular….osteoarthritis [5].” Is misplaced and should be moved earlier, at line 46, soon after citation [4 ]
Line 85-92. This part belongs to Materials and Methods section, not to Introduction. The latter section should terminate with hypothesis and aim of the study.
Mat & Methods section
Reading from the abbreviations table at the end of the manuscript, A-PRP stands for autologous PRP. Are the other PRPs non-autologous? What is the difference between this and others?
Lines 97-125 please avoid repetition in describing the two parts of the study. The list of 8 different PRPs in parentheses is repeated multiple times in the manuscript, which is unnecessary
What is the difference in preparation protocol between PRP-G and PRP-S? How are the PRPs activated in these protocols?
In M& M it should be clearly described the method/protocol of preparation of any type of PRP for any of the kits used, not just label them with different names. It is suggested to add a table, specifying for each PRP type, at least the centrifugation parameters like single/double centrifugation, rpm speed and/or RCF, duration of centrifugation, temperature, automatic/non-automatic method, product yield (if known), the anticoagulant used, the tube material (glass/plastic). Furthermore, as the topic of the study is platelet concentrates, it would be valuable to the readers to indicating also the platelet concentration ratio achieved respect to baseline for each preparation protocol.
Were the 8 PRPs activated or not after preparation? If yes, how?
The P-PRP (leukocyte-poor PRP) is traditionally used as a synonym for the PRGF-Endoret (plasma rich in growth factors – endogenous regenerative therapy), a commercial system produced by BTI Biotechnology Institute, characterized by absence of leukocytes in the final product, obtained by a single centrifugation. Such system has a wide supporting literature in many different medical fields. Is the “P-PRP” used in this study comparable to that P-PRP?
In Table 2 the units superscripts should be checked.
The blood counts must be provided also for the single healthy donor. From data on height and weight, the BMI of this donor was over 35, which is significantly higher than BMI of the 6 participants to the first part of the study (26±5). Since being overweight can have a relevant impact on the metabolism, there is a serious risk that all the metabolomic study could be affected, reducing the transferability of results. Please comment on this in the discussion.
Line 198-217 all this part must be deleted! The authors forgot to remove it from the template…
Results
The first two sentences of Results section do not belong to Results section.
Figure 2 and Fig 6 both represent “VIP score plots presenting the top metabolites responsible for the separation between different PRP preparation methods.” It can be noted that there is not even a single metabolite present in both plots. They represent completely different situations. What is the explanation for such divergence? Same for 3/7, 4/8. It is briefly mentioned in the discussion, but a more clear explanation is required.
However, the clear differences in metabolic fingerprints and heatmaps resulting from different PRP preparations are really striking and very interesting. It is a pity that they cannot be validated because they derived from the blood of a single overweight donor.
Discussion
The first period of discussion, with the list of the main growth factors and the therapeutic potential of PRP is somehow a repetition of the introduction.
Line 324-325. Instead of “therapeutic efficacy” it would be more suitable and prudent to put “biological activity”. Absence of standardization cannot be considered responsible for therapeutic efficacy or inefficacy.
Line 355-356. The authors keep on underlining the differences deriving from different activation methods, but didn’t explain what the different activations consist of.
Line 368-369. This is a speculation, it is suggested to tone down the sentence.
Indeed, most of the Discussion is speculation, since it is based on results deriving from a single subject (having a BMI not representative of the average population). It is surprising that this fact has not be added to the study limitations.
Author Response
Response to Reviewer #2's Comments
Comment 1: The abstract should not start with method, but one-two sentences of background and aim should introduce the methods.
Response: Thank you for this valuable suggestion. The abstract has been rewritten accordingly to include a brief background and aim before describing the methods. The resuts section of the abstract was also expanded.
Comment 2: Please avoid “We”, use impersonal form throughout the paper
Response: Thank you for your comment. The manuscript has been revised to avoid the use of “we,” and an impersonal form has been adopted throughout the text as suggested.
Comment 3: Acronyms should not be introduced without prior explanation (in the abstract is not necessary to list the acronyms, while in the main text reference to the abbreviations list it should be indicated). In the abstract just indicate that PRP-G and PRP-S are activated and prepared with two different kits.
Response: Acronyms were removed from the abstract.
Comment 4: Please note that in the abstract, as well as in the main text, it is not indicated what the difference between the two kits is.
Response: The methodological differences between the two kits used to prepare PRP-G and PRP-S have now been clarified and included in the main text.
PRP-G was obtained using the Quantix IDRIA G kit, which contains ACD-A anticoagulant, sodium citrate, and a thixotropic separation gel within vacuum tubes. The tubes were first centrifuged at 4150 RPM for 3 minutes at +22°C to properly stratify the separation gel before blood collection. After drawing venous blood, a second centrifugation was performed at 3100 RPM for 8 minutes at +22°C. The upper yellowish phase (platelet-rich plasma) was then collected and passed through a built-in filtration system to obtain the activated PRP-G samples.
PRP-S was prepared using the Quantix IDRIA S kit, which includes vacuum tubes containing glass beads. After 28 mL of venous blood was drawn into the tubes, they were gently inverted 10 times and left at room temperature for 30 minutes to allow full interaction between the blood and the beads. Subsequently, the samples were centrifuged at 4000 RPM for 10 minutes. The upper serum phase, enriched with autologous cytokines and platelets, was collected as PRP-S.
Comment 5: In spite of the interesting methodological approach, the sentence “Our results clearly show that the choice of preparation protocol, rather than donor-related factors, is the primary determinant of PRP’s biochemical composition and potential clinical utility.” cannot be supported with results deriving from tests performed on the blood of a SINGLE donor. This comment applies not only to the abstract.
Response: Thank you very much for this important observation. The primary reason for using a single donor in the second part of the study was to eliminate inter-donor variability and isolate the effects of the preparation methods themselves. This rationale has already been stated in the abstract and is further clarified in the main text. To avoid overinterpretation, the relevant conclusions has been rephrased accordingly.
Comment 6: Line 38-40. Why only TGF-beta acronym was mentioned in the listed six growth factors?
Response: As abbreviations were not used for the other growth factors and the term TGF-beta does not appear again in the text, its acronym has been removed to ensure consistency. The sentence has been revised accordingly.
Comment 7: Line 49-50. The sentence “Multiple intra-articular….osteoarthritis [5].” Is misplaced and should be moved earlier, at line 46, soon after citation [4 ]
Response: Thank you for the suggestion. The sentence has been repositioned as recommended to improve the logical flow of the paragraph.
Comment 8: Line 85-92. This part belongs to Materials and Methods section, not to Introduction. The latter section should terminate with hypothesis and aim of the study.
Response: The suggested revision has been made accordingly.
Comment 9: Reading from the abbreviations table at the end of the manuscript, A-PRP stands for autologous PRP. Are the other PRPs non-autologous? What is the difference between this and others?
Response: All PRP preparations used in this study are autologous. The term “A-PRP” refers to “activated PRP” based on a specific commercial system, and the designation follows the manufacturer’s terminology. The description in the abbreviations table has been revised accordingly.
Comment 10: Lines 97-125 please avoid repetition in describing the two parts of the study. The list of 8 different PRPs in parentheses is repeated multiple times in the manuscript, which is unnecessary
Response: The repetitive descriptions of the two parts of the study have been revised for clarity and conciseness. Additionally, the list of eight PRP types in parentheses has been removed from repeated instances throughout the manuscript to improve readability.
Comment 11: What is the difference in preparation protocol between PRP-G and PRP-S? How are the PRPs activated in these protocols?
Response: As indicated in your previous remark regarding the abstract, the differences between the two kits have now been clarified and added to both the abstract and the main text. The preparation procedures are as follows:
PRP-G was obtained using the Quantix IDRIA G kit, which contains ACD-A anticoagulant, sodium citrate, and a thixotropic separation gel within vacuum tubes. The tubes were first centrifuged at 4150 RPM for 3 minutes at +22°C to properly stratify the separation gel before blood collection. After drawing venous blood, a second centrifugation was performed at 3100 RPM for 8 minutes at +22°C. The upper yellowish phase (platelet-rich plasma) was then collected and passed through a built-in filtration system to obtain the activated PRP-G samples.
PRP-S was prepared using the Quantix IDRIA S kit, which includes vacuum tubes containing glass beads. After 28 mL of venous blood was drawn into the tubes, they were gently inverted 10 times and left at room temperature for 30 minutes to allow full interaction between the blood and the beads. Subsequently, the samples were centrifuged at 4000 RPM for 10 minutes. The upper serum phase, enriched with autologous cytokines and platelets, was collected as PRP-S.
Comment 12: In M& M it should be clearly described the method/protocol of preparation of any type of PRP for any of the kits used, not just label them with different names. It is suggested to add a table, specifying for each PRP type, at least the centrifugation parameters like single/double centrifugation, rpm speed and/or RCF, duration of centrifugation, temperature, automatic/non-automatic method, product yield (if known), the anticoagulant used, the tube material (glass/plastic). Furthermore, as the topic of the study is platelet concentrates, it would be valuable to the readers to indicating also the platelet concentration ratio achieved respect to baseline for each preparation protocol.
Response: Thank you for your valuable suggestion. In response to your comment, a detailed table has been prepared outlining the specific methodological parameters for each PRP preparation method used in the study. This table includes information such as the type of centrifugation (single/double), RPM values, centrifugation duration, product yield and tube material (glass/plastic). Additionally, where available, the platelet concentration ratio achieved for each PRP type relative to the baseline is also provided. The table is presented below.
|
PRP Type |
Kit Name |
Centrifugation (Single/Double Spin) |
RPM (revolutions per minute) |
Duration (min) |
Yield (mL) |
Tube Material |
Activated |
Leukocyte Content |
Platelet Content |
|
PRP-P |
MSK Premium PRP |
Single |
3500 |
8 |
2–3 |
Gel-based plastic |
No |
0.04x103/μL |
168 |
|
A-PRP |
MSK Activated PRP |
Single |
3000 |
6 |
2–3 |
Gel-based plastic |
Yes |
0.03x103/μL |
153 |
|
B7-PRP |
MSK Biotin PRP |
Single |
3000 |
6 |
4–5 |
Gel-based plastic |
No |
0.01x103/μL |
56 |
|
P-PRP |
Neogenesis |
Single |
3100 |
10 |
~3 |
Plastic |
No |
0 |
19 |
|
L-PRP |
Neogenesis |
Single |
3100 |
10 |
~4 |
Plastic |
No |
1.44 x103/μL |
50 |
|
C P-PRP |
CellPhi |
Single |
3400 |
10 |
5.5 |
Plastic |
No |
0 |
39 |
|
C L-PRP |
CellPhi |
Single |
3400 |
10 |
5.5 |
Plastic |
No |
1.67 x103/μL |
320 |
|
TFF-PRP |
Manson HA-PRP |
Single |
3500 |
8 |
2–3 |
Gel-based plastic |
No |
0.56 x103/μL |
130 |
|
PRP-G |
Quantix IDRIA G |
Double |
4150 / 3100 |
3 / 8 |
~13 |
Gel-based plastic |
Yes |
NA |
NA |
|
PRP-S |
Quantix IDRIA S |
Single |
4000 |
10 |
~13 |
Glass beads based plastic |
Yes |
NA |
NA |
Comment 13: Were the 8 PRPs activated or not after preparation? If yes, how?
Response: A-PRP and B7-PRP were activated. No further exogenous activation steps (e.g., calcium chloride or thrombin) were applied to any of the PRP samples.
Comment 14: The P-PRP (leukocyte-poor PRP) is traditionally used as a synonym for the PRGF-Endoret (plasma rich in growth factors – endogenous regenerative therapy), a commercial system produced by BTI Biotechnology Institute, characterized by absence of leukocytes in the final product, obtained by a single centrifugation. Such system has a wide supporting literature in many different medical fields. Is the “P-PRP” used in this study comparable to that P-PRP?
Response: Thank you for your comment. While PRGF-Endoret is commonly described as a leukocyte-poor PRP obtained by single centrifugation, in practice it involves a second centrifugation step following the addition of calcium gluconate to the platelet-poor plasma fraction. In this study, the P-PRP preparation was obtained through a different commercial system using a single centrifugation without any exogenous activators. Therefore, although both preparations are leukocyte-poor, the protocol used in this study is not directly comparable to the PRGF-Endoret system.
Comment 15: In Table 2 the units superscripts should be checked.
Response: The superscripts in the units presented in Table 2 have been reviewed and corrected.
Comment 16: The blood counts must be provided also for the single healthy donor. From data on height and weight, the BMI of this donor was over 35, which is significantly higher than BMI of the 6 participants to the first part of the study (26±5). Since being overweight can have a relevant impact on the metabolism, there is a serious risk that all the metabolomic study could be affected, reducing the transferability of results. Please comment on this in the discussion.
Response: Thank you for raising this important issue. In clinical practice, we almost always treat patients over the age of 60 who are often overweight or obese. For this reason, we preferred to harvest platelets and leukocytes from a representative individual. We acknowledge that the metabolic profile of this donor may have influenced the results; however, the primary aim of this part of the study was to eliminate inter-individual variability. We agree that further research focusing on metabolic differences related to obesity is important and warranted.
Comment 17: Line 198-217 all this part must be deleted! The authors forgot to remove it from the template…
Response: Thank you for pointing this out. The template text has now been removed from the manuscript.
Comment 18: The first two sentences of Results section do not belong to Results section.
Response: The first two sentences of the Results section have been removed, and the section now begins directly with the presentation of the relevant findings.
Comment 19: Figure 2 and Fig 6 both represent “VIP score plots presenting the top metabolites responsible for the separation between different PRP preparation methods.” It can be noted that there is not even a single metabolite present in both plots. They represent completely different situations. What is the explanation for such divergence? Same for 3/7, 4/8. It is briefly mentioned in the discussion, but a more clear explanation is required.
Response: We appreciate the opportunity to elaborate further on this important point. First, it is essential to note that due to the nature of untargeted metabolomics, it is not technically possible to detect the entire human metabolome (~15,000 endogenous metabolites) in a single LC-MS injection. In practice, a single untargeted LC-MS run can detect between 3,000–5,000 features (peaks), of which only 700–1,000 can be annotated as metabolites after stringent quality filtering and spectral matching. Even fewer are consistently detected across all samples and can be quantitatively compared. Therefore, the absence of overlapping metabolites between different VIP plots or heatmaps does not imply inconsistency or error—it is a recognized limitation of the field.
Secondly, the datasets for Figures 2–4 (multi-donor, PRP-G vs. PRP-S) and Figures 6–8 (single-donor, eight PRP types) were acquired and processed at different time points. As such, they were not included in the same batch or analytical dataset, to avoid introducing cross-batch bias and technical variability (e.g., due to column performance, ion source fluctuations, or instrument drift). In metabolomics, it is analytically inappropriate to directly compare datasets generated in separate experimental runs unless strict batch normalization is applied (e.g., using pooled QC samples or reference standards across both runs). Since that was not feasible here, we chose to analyze and interpret each dataset independently, in line with accepted best practices. It is also important to recognize that many metabolic pathways contain multiple metabolites, and any given PRP formulation may shift the pathway by altering the abundance of only a subset of detectable compounds. This means that while the pathway enrichment may appear similar, the actual VIP metabolites driving that enrichment may differ. Additionally, metabolites that do not show significant changes (i.e., no VIP or fold-change threshold crossing) are not plotted, although they may in fact overlap across datasets.
Finally, we emphasize that this was an untargeted discovery-based study, not a targeted panel-based quantification. If we had selected specific pathways (e.g., energy metabolism, steroid biosynthesis) and developed a targeted assay, the overlap between key metabolites across different PRP types might have been greater. However, this untargeted design provides a broader view of PRP metabolic diversity. We have now added a paragraph in the revised Limitations and Future Perspectives section to clarify this point.
Comment 20: However, the clear differences in metabolic fingerprints and heatmaps resulting from different PRP preparations are really striking and very interesting. It is a pity that they cannot be validated because they derived from the blood of a single overweight donor.
Response:** Thank you very much for this important observation. We completely agree that using a larger and more diverse sample set would strengthen the generalizability and external validation of the results.
However, we respectfully would like to point out that this limitation has already been acknowledged and discussed in the original version of the manuscript (see “Limitations and Future Perspectives” section). That said, we fully understand the need for further clarity and have now significantly expanded and improved this section in the revised manuscript to explain both the limitations and advantages of our intra-individual design more explicitly. Additionally, we would like to draw your attention to Figure 3, which compares PRP-G and PRP-S prepared from six different donors. The heatmap shows strong clustering within each PRP type and clear separation between the two, despite expected inter-individual differences. This supports our rationale that PRP preparation method, rather than donor variability, is the primary determinant of metabolomic profile differences. Therefore, while we acknowledge that a single-donor design limits population-level generalizability, it also offers a methodological advantage by eliminating donor-based confounding. All eight PRP formulations in the intra-individual comparison were derived from the same blood, meaning the differences observed are exclusively kit-related. In summary, we have now clarified these points and emphasized both the limitation and the rationale in the revised “Limitations and Future Perspectives” section.
Comment 21: The first period of discussion, with the list of the main growth factors and the therapeutic potential of PRP is somehow a repetition of the introduction.
Response: As suggested, the initial paragraph of the Discussion section, which included the list of major growth factors and general therapeutic potential of PRP, has been removed to avoid redundancy with the Introduction.
Comment 22: Line 324-325. Instead of “therapeutic efficacy” it would be more suitable and prudent to put “biological activity”. Absence of standardization cannot be considered responsible for therapeutic efficacy or inefficacy.
Response: As per your suggestion, the term “therapeutic efficacy” in lines 324–325 has been replaced with “biological activity”; in addition, similar expressions throughout the manuscript have also been revised accordingly.
Comment 23: Line 355-356. The authors keep on underlining the differences deriving from different activation methods, but didn’t explain what the different activations consist of.
Response: The activation methods have now been clearly described in the revised version of the manuscript, including the procedural details for A-PRP, PRP-G and PRP-S, which were activated as part of their respective commercial kit protocols.
Comment 24: Line 368-369. This is a speculation, it is suggested to tone down the sentence.
Response: We agree that the sentence contained a speculative and inferential expression, and we have therefore revised it to adopt language that is more consistent with the data presented.
Comment 25: Indeed, most of the Discussion is speculation, since it is based on results deriving from a single subject (having a BMI not representative of the average population). It is surprising that this fact has not be added to the study limitations.
Response: We agree that relying on data from a single subject limits the generalizability of the findings. However, as noted in our previous responses, this was a deliberate design choice aimed at eliminating inter-individual variability in order to isolate the effects of PRP preparation methods on metabolomic profiles. We have now explicitly addressed this point in the revised “Limitations and Future Perspectives” section, highlighting both the limitations and the rationale behind using a single, non-representative donor.
Thank you again for your valuable feedback. These revisions have significantly improved our manuscript, and we are confident that it now presents our findings more effectively. In addition, we have strengthened the Discussion section and incorporated new references to better contextualize our results within the existing literature.

Round 2
Reviewer 1 Report
Comments and Suggestions for Authors
For the preparation of platelet-rich plasma (PRP), this study predominantly opted for single-centrifugation methods. In fact, a substantial number of studies have employed double-centrifugation techniques for PRP preparation.
Author Response
Dear Reviewer,
Thank you very much for your attention, The platelet content column was mistakenly written wrongly during our last upload. We only performed pre and post leukocyte, red blood cell and platelet count in the L-PRP group. The leukocyte count before and after the PRP preparation remained almost unchanged. The number of red blood cells decreased significantly. The thrombocyte count increased from 230.000 μL to 447.250 μL, which confirmed the concentration of the thrombocytes. This information was integrated into Table 2 as a footnote.
Your input significantly helped us to walk out from a major mistake. We once again thank to your attention and contribution.
Sincere regards,
Feza Korkusuz MD
Reviewer 2 Report
Comments and Suggestions for Authors
Dear Authors, thank you for addressing my previous comments. The paper is much clear and improved now, and I thinks it adds an original piece of knowledge in the field of platelet concentrates.
I only have a minor point. In the added table 2 the columns "Yield" and "Platelet content" should be fixed. I am not sure that the unit for yield is mL, as the final volume depends on the initial volume of blood drawn. Does the numbers represent the final/initial platelet concentration ratio? Please clarify. For the final column, the unit is missing. Is it the same as "Leukocyte content" (103/microliter)? In this case, some PRPs have a really low platelet content! 19, 39, 50, 56... How could they be defined platelet concentrates?
Author Response

(The authors gave the same response as above.)
